# Artificial Sweeteners Negatively Regulate Pathogenic Characteristics of Two Model Gut Bacteria, *E. coli* and *E. faecalis*

**DOI:** 10.3390/ijms22105228

**Published:** 2021-05-15

**Authors:** Aparna Shil, Havovi Chichger

**Affiliations:** 1Biomedical Research Group, School of Life Sciences, East Road, Cambridge CB1 1PT, UK; aparnashil12@gmail.com; 2School of Life Sciences, Anglia Ruskin University, Cambridge CB1 1PT, UK

**Keywords:** artificial sweeteners, microbiota, in vitro models, gut bacteria

## Abstract

Artificial sweeteners (AS) are synthetic sugar substitutes that are commonly consumed in the diet. Recent studies have indicated considerable health risks which links the consumption of AS with metabolic derangements and gut microbiota perturbations. Despite these studies, there is still limited data on how AS impacts the commensal microbiota to cause pathogenicity. The present study sought to investigate the role of commonly consumed AS on gut bacterial pathogenicity and gut epithelium-microbiota interactions, using models of microbiota (*Escherichia coli* NCTC10418 and *Enterococcus faecalis* ATCC19433) and the intestinal epithelium (Caco-2 cells). Model gut bacteria were exposed to different concentrations of the AS saccharin, sucralose, and aspartame, and their pathogenicity and changes in interactions with Caco-2 cells were measured using in vitro studies. Findings show that sweeteners differentially increase the ability of bacteria to form a biofilm. Co-culture with human intestinal epithelial cells shows an increase in the ability of model gut bacteria to adhere to, invade and kill the host epithelium. The pan-sweet taste inhibitor, zinc sulphate, effectively blocked these negative impacts. Since AS consumption in the diet continues to increase, understanding how this food additive affects gut microbiota and how these damaging effects can be ameliorated is vital.

## 1. Introduction

There are more than 100 trillion microbes in the gut which encode hundredfold more unique genes than the human genome [1,2]. There is a continued, concerted effort to further understand this, for example, with the METAgenomics of the Human Intestinal Tract collaborative project which investigates the contribution of the microbiome in providing unique protein coding genes at levels which are 360 times greater than their host [3,4]. The gut microbiome is in close proximity to the intestinal epithelium and, as such, epithelial cells of the intestine are exposed to diverse antigenic substances from both dietary components and gut microbiota activities. Indeed, the importance of commensal microbiota for intestinal epithelial function has already been demonstrated in germ-free mice which display a significant reduction in mucosal layer thickness and antimicrobial product levels [5,6,7]. Microbial metabolites from the gut microbiota, such as short-chain fatty acids (SCFA), indole, and lactate have also been shown to regulate intestinal barrier function [8,9]. Higher SCFA concentrations have been linked with increased gut dysbiosis, gut permeability and are indicators of various metabolic diseases, whilst improved microbial diversity showed opposite correlation with these outcomes [10]. Therefore, the gut microbiota, and associated metabolic products, play a significant role in intestinal health and wellbeing of the host.

There is a symbiotic relationship between microbiota and host where metabolic products of gut microbiota, such as B group vitamins and vitamin K, provides essential support for human growth and development [11]. However, diet is also one of the most important factors that shapes gut microbiota. Studies have demonstrated that a diet which is low in fat and animal protein but rich in complex carbohydrates provides the host with a more diverse, and therefore “healthy”, gut microbiome [12]. Adult healthy volunteers who consumed a large amount of animal-based diet displayed a shift in bacterial phyla composition associated with increased presence of the bile-tolerant enterotypes (*Alistipes*, *Bilophila,* and *Bacteroides*) and decreased the quantities of Firmicutes that metabolize plant polysaccharides (*Roseburia*, *Eubacterium rectale,* and *Ruminococcus bromii*) in the gut microbiota [13]. Similar observations were found comparing the intestinal microbiota of children in a rural village in Africa with an urban area in Europe, where composition of diet is dramatically impacted by the Western diet [14]. Whilst these studies demonstrate a change in bacterial phyla following long-term changes in diet, short-term exposure of healthy individuals to an energy-rich diet have been found to affect the function of the gut microbiota with negative changes observed in bacterial secretion system, protein export, and lipoic acid metabolism [12]. Interestingly, it is not only key food groups which impact on the gut microbiota. Studies demonstrate that additives, typically used to improve the taste, appearance and longevity of foods and drinks, also have an impact on the composition and function of gut bacteria [5,15]. For example, the FDA-approved dietary emulsifiers carboxymethyl cellulase and polysorbate 80, when administered to mice at acceptable daily intake levels, increase the number of mucolytic bacteria, and decrease the number of *Bacteroidetes* in the gut microbiota [5]. These changes result in a reduced colonic mucin layer and, as such, several studies have demonstrated the association between emulsifier exposure and metabolic syndrome development [5]. Therefore, there is a well-established link between dietary additives, dysregulation of microbiota, and subsequent impact on gut health.

In recent years, artificial sweeteners have become popular as a non-caloric additive to sweeten foods and drinks. Artificial sweeteners, such as sucralose and aspartame, provide the sweet taste in low-calorie foods, which have increased their popularity worldwide [16]. As these sweeteners are cheap, easily available, and result in enhanced food flavour, they have been incorporated into many food products and beverages, as well as pharmaceutical products. Epidemiological studies have evidenced the beneficial role of sweeteners in weight loss and for people suffering from glucose intolerance and type 2 diabetes mellitus [16,17,18]; there are, however, studies which indicate opposing results. Using animal and human studies, artificial sweetener consumption has been linked with conditions leading to metabolic disease development [15]. Indeed, sweeteners were reported to induce glucose intolerance by altering the composition and function of the gut microbiota. In mice, sweetener-intake was linked with dysbiosis leading the host to be prone to symptoms related to metabolic disease. These symptoms were abrogated by treatment with the antibiotics, ciprofloxacin, metronidazole, and vancomycin, which affected the commensal microbiota and ameliorated the metabolic disease symptoms [14]. Further studies have confirmed that aspartame exposure, over 8 weeks, increases fasting glucose levels and insulin intolerance in rats [18]. These studies show that perturbed gut microbiota occurred, in response to sweetener treatment, with an increase in abundance of *Enterobacteriaceae* and *Clostridium leptum*. In a human study on 4-day food intake, the relationship between aspartame and acesulfame potassium intake and microbiota was demonstrated; no differences in the abundance and genetic composition of bacteria were noted between the artificial sweetener consumers and non-consumers, however, a significant difference in microbial diversity was observed [19]. Bian et al. demonstrated that numerous pro-inflammatory mediators were potentially produced by gut bacteria following the consumption of sweeteners in the diet, which is associated with other metabolic disease conditions like diabetes and obesity [20,21]. Interestingly, in gut epithelial cells, our recent studies demonstrate that exposure to artificial sweeteners increases apoptosis and permeability across the intestinal epithelium associated with inflammatory gut leak [22]. Despite controversy in the field, there is strong evidence that without changing the bacterial composition, artificial sweeteners in the diet cause changes in bacterial diversity, and potentially pathogenicity, which is likely to exert a negative impact to the host. However, how artificial sweeteners affect symbiotic bacteria and contribute to pathogenicity remained veiled yet.

*Escherichia coli* (*E.coli)* is the most frequent facultative anaerobic Gram-negative bacterium inhabiting the human GI tract and is a versatile pathogen whilst *Enterococcus faecalis* (*E. faecalis*) is a facultative anaerobic, Gram-positive coccus that inhabits the human GI tract [23,24]. Both *E. coli* and *E. faecalis* are commensal, as well as pathogens, and both colonise immediately after birth so are good representatives of their corresponding phyla [25]. From a technical perspective, both species can be isolated easily, grown, and maintained in the laboratory and are frequently used as potential human faecal indicators. We therefore used *E. coli* and *E. faecalis* as models to test our hypothesis that artificial sweeteners, at physiologically achievable levels in the small intestine [22,26], negatively impact bacteria in the gut microbiome. In the gut environment, both human epithelial cells and the microbiota are exposed to various concentrations of artificial sweeteners when consumed in the diet. We sought to understand the effect of commonly consumed artificial sweeteners, saccharin, sucralose, and aspartame, on two model gut bacteria (*E. coli* and *E. faecalis*) to gain an insight into the potentially pathogenic mechanisms through which sweeteners could impact the microbiota.

## 2. Results

### 2.1. Only the Artificial Sweetener Saccharin Affects E. coli Model Gut Bacteria Growth at High Concentrations

The effect of artificial sweeteners on *E. coli* and *E. faecalis* growth in planktonic culture was measured every 12-h, upon exposure to varying concentrations of artificial sweetener (saccharin, sucralose, and aspartame), for 4 days. Experiments with *E. coli* showed no significant change in normalised growth in response to sucralose or aspartame exposure at any time point or concentration (Figure 1b or c). In contrast, exposure to 1000 µM saccharin significantly reduced *E. coli* growth between 48–84 h, however lower concentrations of saccharin had no impact on *E. coli* growth (Figure 1a). Experiments with *E. faecalis* demonstrated no significant effect of either saccharin, sucralose, or aspartame, on normalised bacteria cell growth, at any time point or concentration (Figure 1d–f).

### 2.2. Artificial Sweeteners Differentially Increase Biofilm Formation, but Not Haemolytic Activity, in the Two Model Gut Bacteria

The ability for artificial sweeteners to regulate the pathogenicity of model gut bacteria was studied by measuring biofilm formation and haemolytic activity assays. Experiments were performed with *E. coli* and *E. faecalis* exposed to saccharin, sucralose, or aspartame at the physiological concentration of 100 µM; the sweeteners did not impact bacteria cell growth, except as seen with *E. coli* and saccharin at 1000 µM (Figure 1a). Biofilm formation of *E. coli* was significantly increased, compared to the vehicle control, when bacteria were exposed to artificial sweeteners saccharin, sucralose, and aspartame (Figure 2a). Whilst biofilm formation of *E. faecalis* was also increased with all three artificial sweeteners, only aspartame exposure caused a significant increase in the ability of *E. faecalis* to form a biofilm (Figure 2b). In contrast, the haemolysis assay indicated that artificial sweeteners saccharin, sucralose, and aspartame had no effect on qualitative (Figure 2c (i)) or semi-quantitative (Figure 2c (ii)) haemolysin production of either *E. coli* or *E. faecalis*, in contrast to the positive control, the haemolytic bacteria *S. aureus*.

### 2.3. Artificial Sweeteners Significantly Disrupt the Interaction between Model Gut Bacterial and Intestinal Epithelial Cells

To further understand the physiological effect of artificial sweeteners on model gut bacteria, *E. coli* and *E. faecalis*, the next studies used a co-culture system with the intestinal epithelial cell line, Caco-2 cells. Studies measured the effect of saccharin, sucralose, and aspartame in regulating the ability of model gut bacteria to adhere to, invade, and kill intestinal epithelial cells.

Intact *E. coli* or *E. faecalis*, which were pre-exposed to artificial sweeteners, were incubated with Caco-2 cells to establish adhesion ability of the model gut bacteria. All three artificial sweeteners studied, saccharin, sucralose, and aspartame, significantly increased adhesion of both *E. coli* and *E. faecalis* to intestinal epithelial cells (Figure 3a,d). Interestingly, a more dramatic fold-increase in bacterial adhesion to Caco-2 cells was observed with *E. faecalis* compared to *E. coli* for saccharin (*E. coli* 2.3 ± 0.4 versus *E. faecalis* 5.2 ± 2.1), sucralose (*E. coli* 2.0 ± 0.3 versus *E. faecalis* 5.4 ± 1.8), and aspartame (*E. coli* 2.9 ± 0.7 versus *E. faecalis* 6.6 ± 1.9).

The ability of model gut bacteria, *E. coli* and *E. faecalis*, to invade Caco-2 cells was measured following exposure to artificial sweeteners. Sucralose and aspartame exposure significantly increased the invasion index of both *E. coli* and *E. faecalis* (Figure 3b,e). In contrast, incubation with saccharin had a significant effect on the invasive ability of *E. faecalis* (Figure 3e) but not *E. coli* (Figure 3b). Soluble bacterial factors released from *E. coli* or *E. faecalis* exposed to artificial sweeteners were incubated with Caco-2 cells and cytotoxicity was measured. Soluble factors secreted from *E. coli* exposed to saccharin and sucralose, but not aspartame, induced a small but significant reduction in Caco-2 cell viability (Figure 3c). In contrast, *E. faecalis* exposed to sucralose and aspartame, but not saccharin, released soluble factors which significantly lowered Caco-2 cell viability (Figure 3f).

### 2.4. Artificial Sweeteners Impact Model Gut Bacteria through a Taste Sensing Mechanism

Our final experiments sought to understand how artificial sweeteners regulated the model gut bacteria pathogenic functions studied: biofilm formation, adhesion and invasion ability, and cytotoxicity. Given the lack of published literature in the field, there is no specific taste receptor established in bacteria. The pan taste inhibitor, zinc sulphate, was used to establish the potential of a taste sensing mechanism in *E. coli* and *E. faecalis* [27]. At a range of concentrations, zinc sulphate has no impact on growth of either model gut bacteria (Figure 4a). Zinc sulphate also had no impact on biofilm formation, adhesive or invasive ability of either bacteria in the absence of sweeteners (vehicle-treated *E. coli* or *E. faecalis*) (Figure 4b–g). In *E. coli*, zinc sulphate significantly reduced sweetener-induced biofilm formation (Figure 4b) and adhesive ability (Figure 4c). Furthermore, aspartame- and sucralose-mediated increase in the ability of *E. coli* to invade Caco-2 cells was attenuated by zinc sulphate (Figure 4d). In *E. faecalis*, zinc sulphate attenuated biofilm formation induced by aspartame (Figure 4e) and all sweetener-induced effects on bacterial adhesion and invasion of Caco-2 cells (Figure 4f,g). The cytotoxic effect on Caco-2 cells, induced by saccharin and sucralose exposure with *E. coli* (Figure 3c) and sucralose and aspartame exposure with *E. faecalis* (Figure 3f), was also blocked by exposure to zinc sulphate (data not shown).

Taken together, findings demonstrate the effect of artificial sweeteners, saccharin, sucralose, and aspartame, in increasing the pathogenicity of model gut bacteria, *E. coli* and *E. faecalis,* through a taste-dependent pathway. Further studies are needed to understand the molecular mechanisms responsible for these pathogenic effects with the aim of reducing the negative impact of sweeteners on gut health.

## 3. Discussion

Artificial sweeteners are commonly consumed in the diet as an aid for weight loss however there is still controversy on the potential benefits versus detrimental effects of these compounds on gut health. Previous studies have demonstrated the impact of sweeteners on diversity of gut microbiota [15,19,28]. These studies indicate an increase in bacteria, such as Enterobacteriaceae, of which *E. coli* is one, however this has been studied in whole microbiome rather than at the individual bacteria or molecular level. In the present study, we use an in vitro model of the gut and the microbiota to investigate the impact of sweeteners on potential changes in pathogenicity. Our studies demonstrate that saccharin, sucralose, and aspartame at the physiological concentration of 100 µM, which could be easily achieved in the diet [15,22], differentially increase biofilm formation as well as the ability of bacteria to adhere to, invade and kill mammalian gut epithelial cells. These findings indicate that saccharin, sucralose, and aspartame all promote pathogenic changes in two model gut bacteria, *E. coli* and *E. faecalis*, which could worsen the effect of consuming artificial sweeteners in the diet on gut health.

Bacterial growth is one of the most well-studied characteristics of metabolism. Planktonic growth of *E. coli* and *E. faecalis* showed little effect from exposure to artificial sweeteners at a range of concentrations. The exception to this is saccharin at 1 mM concentration which caused bacteriostatic effects in *E. coli* from 48 to 84 h. These findings differ from those in the literature where Wang et al. established that saccharin, sucralose, and acesulfame K have bacteriostatic effects on *E. coli* HB101 and K-12 strains [29]. Importantly, this study was performed with supraphysiological sweetener concentrations in the 30–80 mM range which may explain the difference in findings [16]. Our studies also demonstrate that all three sweeteners evaluated (saccharin, sucralose, and aspartame) increased biofilm formation of *E. coli* with only aspartame affecting biofilm formation of *E. faecalis.* Bacteria typically transition from planktonic to biofilm as a result of environmental and physiological cues, including cell density, nutrient availability, and cellular stress. Bacteria growing in biofilms are less sensitive to antimicrobial resistance treatment and are more likely to express more virulence factors and exotoxins compared to planktonic cells [30,31,32]. These virulence factors may be related to those causing increased adhesion to and invasion of Caco-2 cells and elevated cytotoxic effects, as we observed in the present study. It is, however, worth noting that we did not note the same pattern of sweetener effect for the different pathogenicity measurements made for each bacterium. For example, in studies with *E. coli* we established that all 3 sweeteners studied cause an increase in biofilm formation and adhesion to Caco-2 cells. However, only sucralose and aspartame affected bacterial invasion and soluble factors from only saccharin- and sucralose-treated *E. coli* decreased Caco-2 cell viability. Therefore, sweeteners have differential effects on pathogenicity in *E. coli* and *E. faecalis*.

There are a range of virulence factors which bacteria can utilise to become pathogenic to a host, such as prevention of complement activation and escape from phagosomes. Some pathogenic *E. coli* have been shown to display a range of virulence factors, such as bundle-forming pilus (BFP), type 1 pili and cytolysin A (ClyA), to cause adherence and invasion of host cells, and the production of cytotoxins which kill host cells [33,34,35]. Similarly, pathogenic *E. faecalis* shows adhesion and invasion of intestinal epithelial cells, via pili and aggregation substances, such as AsaI and glycolipids, and cytotoxicity via secreted factors, such as cytolysin [36,37,38,39]. Other mechanisms of pathogenicity have also been identified. *E. coli* can exert pathogenic effects, such as biofilm formation, through yafK and Fis gene expression, and α- or β-haemolysis, potentially through a ClyA-mediated pathway [40,41]. Similarly, biofilm formation and haemolysis activity have been shown in pathogenic *E. faecalis* by the xdh or Esp genes and β-haemolysin, respectively [42,43,44]. Whilst artificial sweeteners have been shown to affect dysbiosis in the gut microbiota, there are limited mechanistic studies which show pathogenic responses of individual bacteria to sweeteners. In the present study, we demonstrate that the artificial sweeteners saccharin, sucralose, and aspartame, at physiological concentrations, impact on all these pathogenic mechanisms except β-haemolysis. The model bacteria, *E. coli* and *E. faecalis*, are α- and γ-haemolytic, respectively, in normal conditions, however, they can turn into β-haemolytic when pathogenic [44,45]. It is possible that we noted no change in haemolysis because of the in vitro nature of the study or the use of laboratory strains of each bacteria. Indeed, clinical isolates of *E. coli* or *E. faecalis* have been shown to display haemolytic genes, such as hly and ClyA [46,47], which are likely to be lacking from the bacteria we studied. It is also worth noting that model gut bacteria were exposed to artificial sweeteners for 24 h, so only the long-lasting response to the additive was recorded. However, given that artificial sweeteners are consistently present in the diet, in a range of sources from food, drink, and cosmetics, it is likely that the microbiome would be continuously exposed and long-lasting responses are most accurate to study. Further study on the genetic changes of each bacteria, following exposure to saccharin, sucralose, and aspartame, may provide a deeper molecular understanding of the mechanisms regulating their pathogenicity.

In mammals, sweet taste molecules are recognised via the sweet taste receptor hetero- or homo-dimer, T1R2/T1R3 or T1R3/T1R3, respectively [48]. Zinc sulphate is a potent inhibitor of sweet taste sensing which is hypothesised to act through the zinc ion binding to T1R2/T1R3 or T1R3/T1R3 and altering the conformation of the receptor to prevent sweeteners from interacting with it [27]. Our studies demonstrate that zinc sulphate can effectively block the impact of sweeteners on pathogenicity for both *E. coli* and *E. faecalis*. This indicates the presence of a sweet taste sensor in bacteria through which sweeteners can activate pathogenic effects, however, sucralose has a perceived sweet taste intensity which is 2-times higher than saccharin and 3-times higher than aspartame [49]. Our findings demonstrate differential effects of the three sweeteners on biofilm formation, adhesion and invasion, and cytotoxicity in *E. coli* and *E. faecalis*. Therefore, it is likely that these bacteria are responding independently of traditional taste sensing pathways. Indeed, whilst studies indicate the potential for olfactory responses in *Bacillus licheniformis*, there is no literature indicating the presence of a sweet taste receptor or sensor in bacteria which could respond to artificial sweeteners [50]. Instead, there is evidence that sweeteners can cause DNA damage in bacteria, elevate bacterial mutation rate in a dose-dependent manner or ROS production and detoxification, and increase cell membrane permeability [51,52,53,54]. ROS can modulate the quorum sensing ability of bacteria to sense and respond to their environment [55,56] therefore sweeteners may impact pathogenicity of model gut bacteria in the present study through a ROS-dependent pathway.

Bacteria such as *E. faecalis* have been shown to translocate across the intestinal wall, disseminate into the blood stream, and cause septicaemia along with congregation in the mesenteric lymph nodes, liver, and spleen [57,58,59]. In our present study, we demonstrate that saccharin, sucralose, and aspartame increased the ability of model gut bacteria to adhere to and invade intestinal epithelial cells, with the exception of saccharin which has no significant effect on *E. coli* invasion. Furthermore, we and others have previously demonstrated the negative effect of artificial sweeteners, saccharin, sucralose, and aspartame, on intestinal epithelial cell apoptosis and permeability [22,60], thus further increasing the opportunity for bacteria to traverse the gut epithelium and cause septicaemia. However, to date, no studies have been performed to study the link between consumption of artificial sweeteners correlates with incidence of septicaemia.

Globally, sweeteners (acesulfame, saccharin, and sucralose) have been detected in wastewater, surface water, groundwater, and drinking water systems [61,62,63]. More worryingly, artificial sweeteners have been linked to increased horizontal transfer of antimicrobial resistance genes in environmental and clinical settings [52,54]. In addition, increased biofilm formation is believed to cause medical device-associated infections and is closely linked to the antibiotic-resistant bacteria, which is now a widespread public health threat [64,65,66,67]. Understanding the role of sweeteners in regulating biofilm formation, as well as other pathogenic effects on the intestinal epithelium and antimicrobial resistance of bacteria, could have a dramatic impact on public health in a multitude of ways.

## 4. Materials and Methods

### 4.1. Materials

*Enterococcus faecalis* (*E. faecalis,* 19433™) and *Escherichia coli (E. coli,* 10418) were purchased from ATCC (Middlesex, UK) and NCTC (Salisbury, UK), respectively. *Staphylococcus aureus*, for use as positive control for the haemolysis assay, was a kind gift from Dr. Christopher O’Kane (Anglia Ruskin University). Bacterial media and blood agar plates was purchased from Oxoid (ThermoFisher, Hampshire, UK). For bacterial growth curve experiment and biofilm assay, sterile, flat-bottom, non-treated polystyrene 96-well plates were purchased from CytoOne (StarLabs, Milton Keynes, UK). Phosphate Buffered Saline (PBS) was obtained from Gibco (ThermoFisher, Hampshire, UK). Human colon adenocarcinoma cells (Caco-2, 86010202), Eagle’s Minimal Essential Media (EMEM), zinc sulphate, antibiotics, and artificial sweeteners (saccharin, sucralose, and aspartame) were purchased from Sigma-Aldrich (Dorset, UK).

### 4.2. Bacterial and Mammalian Cell Culture

Bacterial cells were grown aseptically at 37 °C on solid media for single colonies, or in liquid media with shaking (150 rpm) for growth measurements. Brain Heart Infusion agar and Nutrient Agar were used as solid, and Brain Heart Infusion and Nutrient broth were used as liquid media for *E. faecalis* and *E. coli*, respectively, as guided by the supplier.

A human intestinal epithelial cell line, Caco-2 cells, was used for the interaction assays, such as adhesion, invasion, and cytotoxicity assays. Monolayers of Caco-2 cells were grown aseptically in EMEM supplemented with 10% foetal bovine serum and 1% antibiotic (1 U/mL penicillin, 1 μg/mL streptomycin) solution at 37 °C in 5% CO_2_.

### 4.3. Growth Curve Determination

A single bacterial colony of *E. coli* or *E. faecalis* was inoculated aseptically into Nutrient broth or Brain-heart infusion, respectively, supplemented with the artificial sweeteners, saccharin, sucralose, and aspartame at ranging concentrations from 0.1 to 1000 µM, or vehicle (double-distilled water (ddH_2_O)) and allowed to grow for up to 4 days. Growth was recorded as absorbance at 600 nm (A_600_) using the Victor^TM^ X3 (Perkin Elmer) and values were normalised to 0 µM at 0 h (as 1).

### 4.4. Biofilm Formation Assay

Biofilm formation of *E. coli* and *E. faecalis* was measured after exposure to artificial sweeteners, saccharin, sucralose, and aspartame (100 µM) using crystal violet biofilm formation assay as described previously [68] with some modifications. A single bacterial colony was inoculated into 10 mL of the corresponding liquid media supplemented with sweetener or vehicle (H_2_O) in presence or absence of zinc sulphate. Absorbance at 600 nm was measured on a Victor^TM^ X3 multiplate reader to ensure equal bacterial cell numbers, and overnight culture was transferred into liquid media (1:200) supplemented with artificial sweeteners. After vortexing, 200 µL was transferred into sterile 96-well plasticware plates and grown aerobically for 48 h at 37 °C. The supernatant was removed, and wells were washed twice with ddH_2_O to remove loosely associated bacteria. Each well was stained with 150 µL 0.1% Gram crystal violet for 20 min at room temperature. After staining, wells were washed with ddH_2_O three times. The retained crystal violet by the biofilms were brought into solution by adding 200 µL 30% acetic acid and incubating at 37 °C for 5 min. The quantitative analysis of biofilm formation was performed by measuring absorbance at 600 nm using the multiplate reader (Victor^TM^ X3). The biofilm forming unit was calculated by dividing the absorbance of crystal violet retained with the absorbance of the total bacterial growth and was normalised to the control (as 1).

### 4.5. Haemolysis Assay Using Blood Agar Plates

Blood Agar plate with 7% Horse Blood was used to investigate the haemolytic properties of *E. coli* and *E. faecalis* after exposure to artificial sweeteners. Bacteria were exposed to the artificial sweeteners, saccharin, sucralose, and aspartame (100 µM) for 24 h, in the presence and absence of zinc sulphate (100 µM) under shaking conditions at 37 °C. Absorbance of bacteria was assessed at 600 nm and bacteria number was normalised prior to plating on blood agar plates. Plates were incubated at 37 °C and haemolysis was assessed at 24 h alongside *Staphylococcus aureus* as a positive control [69]. Images of the plates were taken using UVI-Tec imager (Uvitec Ltd., Cambridge, UK). To perform semi-quantitative analysis of the haemolysis assay, images were assessed using ImageJ 1.51 s (http://imagej.nih.gov/ij, 19 March 2021). For each streak, the area of the bacteria and the haemolysis area on the blood agar plates was measured in relative units and the haemolytic potential was calculated as the haemolysis area minus the bacteria area.

### 4.6. Adhesion Assay

Adhesion of the model gut bacteria to Caco-2 cells following artificial sweetener exposure was measured as previously described [70] with some modifications. Caco-2 cells were seeded on 24-well tissue culture plates (7.5 × 10^4^ cells/well) and incubated in humidified condition (90%) at 37 °C and 5% CO_2_ for 48 h, following exposure to artificial sweeteners for 24 h. Meanwhile, a single colony of *E. coli* and *E. faecalis* was inoculated into 10 mL of respective media supplemented with the artificial sweeteners in the presence or absence of zinc sulphate, or vehicle (ddH_2_O) and incubated overnight at 37 °C with shaking at 150 rpm. Bacteria were then washed twice with 500 µL serum and antibiotic-free EMEM media by centrifuging at 4000 rpm (2683× *g*) for 10 min at 37 °C (accuSpinTM 1R, Fisher Scientific, Thermo Electron Corporation LED GmbH, Osterode, Germany) and re-suspended in EMEM without antibiotics.

Caco-2 cell monolayers were washed twice with 500 µL PBS, and then EMEM (490 µL; without antibiotics) was added to each well. The total number of adherent Caco-2 cells was measured by performing a cell count. Bacterial suspension (10 µL) was added on the Caco-2 cells at a multiplicity of infection (MOI) 1:300 for an infection incubation time of 1 h. After the infection period, the cells were washed twice with 500 µL of sterile PBS, the Caco-2 cells were lysed with 500 µL of 0.5% Triton X-100 by pipetting up and down. The number of viable bacteria was determined by spread-plating serial dilutions of the cell suspension on respective solid media, followed by overnight incubation at 37 °C and counting colony forming units. Bacterial adhesion was expressed as ratio of total bacteria attached per viable Caco-2 cells (normalised to 1). Each assay was performed in triplicate with the successive passage of Caco-2 cells.

### 4.7. Invasion Assay

Caco-2 cells were seeded on 24-well tissue culture plates at a density of 7.5 × 10^4^ cells/well for 36 h followed by exposure to artificial sweeteners for 24 h at 37 °C in humidified condition with 5% CO_2_. The cell monolayer was washed twice with 500 µL sterile PBS and 490 µL fresh EMEM media without antibiotics was added. Bacterial invasion of Caco-2 cells was measured using the antibiotic protection assay previously described [71] with some modifications. Bacteria were exposed to the artificial sweeteners and prepared for infection, as described in Section 4.6. The number of adhered Caco-2 cells that were subjected to bacterial infection was determined by performing a cell count. Caco-2 cell monolayer was infected with bacteria at MOI 1:300 for 1 h at 37 °C. The monolayer was washed once with 500 µL PBS and fresh cell culture medium (500 µL) was added containing 100 µg/mL gentamicin for *E. coli* and 100 µg/mL gentamicin along with 50 µg/mL ampicillin for *E. faecalis* and incubated at 37 °C for 30 min to kill the external-adhered bacteria [71]. The cell monolayer was washed twice with PBS and then lysed with 0.5% Triton X-100 in PBS.

The number of viable colony-forming units were determined by diluting and plating the samples onto solid media and incubating overnight at 37 °C. The results were expressed as the ratio of intracellular bacteria compared with the control (normalized to 1). Each assay was performed in triplicate with the successive passage of Caco-2 cells.

### 4.8. Cytotoxicity Assay

The cytotoxic effect of AS-mediated bacterial metabolites on intestinal epithelial cells was performed following the protocol previously described [72] with modifications, and cell viability was measured by using the Cell Counting Kit-8 (CCK-8), as per manufacturer’s guidelines. Caco-2 cells were grown on 96-well plates (1 × 10^4^ cells/well) and incubated for 48 h at 37 °C in humidified condition with 5% CO_2_. Simultaneously, *E. coli* or *E. faecalis* was grown in 10 mL of respective liquid media supplemented with 100 μM of artificial sweeteners with or without 100 μM/mL zinc sulphate or vehicle for 24 h. The cultures were centrifuged at 4000 rpm (2683× *g*) for 15 min at 4 °C and supernatant was collected, filter-sterilized (0.22 µM membranes; Millipore, USA). 50 μL of the soluble bacterial factors (supernatant) and 50 μL EMEM without antibiotics was added to the pre-washed (with 100 μL of sterile PBS) Caco-2 cell monolayer. Cells were incubated for 24 h followed by measurement of cell viability using CCK-8 reagent assessed as absorbance at 450 nm using a microplate reader (Tecan Sunrise^TM^, Denmark, Switzerland).

### 4.9. Statistical Analysis

All the quantitative data was collected as Excel (Microsoft Office) files. The average of triplicates for each study were analysed as *n* = 1 with *n* = 5–6 for all studies except the haemolysis assay. Data were analysed using GraphPad Prism 7. Statistical analysis was performed using either a one-way ANOVA or a two-way ANOVA, with Tukey multiple comparisons post-hoc test where relevant. Significance was reached when *p* < 0.05.

## Figures and Tables

**Figure 1 ijms-22-05228-f001:**
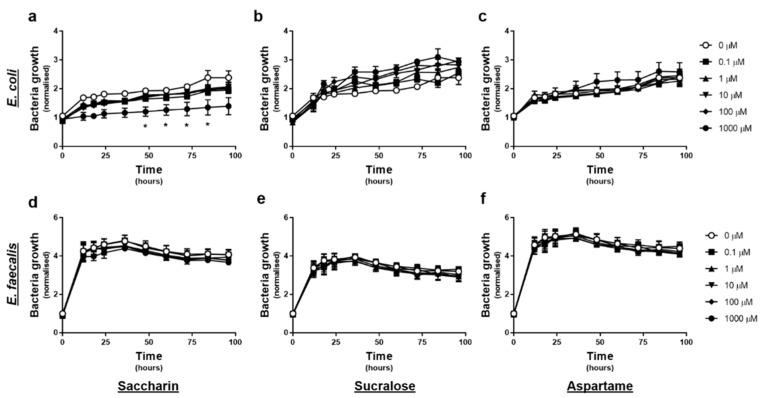
Only the artificial sweetener saccharin affects *E. coli* model gut bacteria growth at high concentrations. *E. coli* (panels **a**–**c**) and *E. faecalis* (panels **d**–**f**) growth was measured, following exposure to the artificial sweeteners (0 to 1000 µM), saccharin (panels **a**,**d**), sucralose (panels **b**,**e**), and aspartame (panels **c**,**f**), for up to 96 h. Specific bacteria for each study are included in underlined text. Data are presented as mean ± standard error mean (S.E.M.). * *p* < 0.05 versus 0 µM at the same time point.

**Figure 2 ijms-22-05228-f002:**
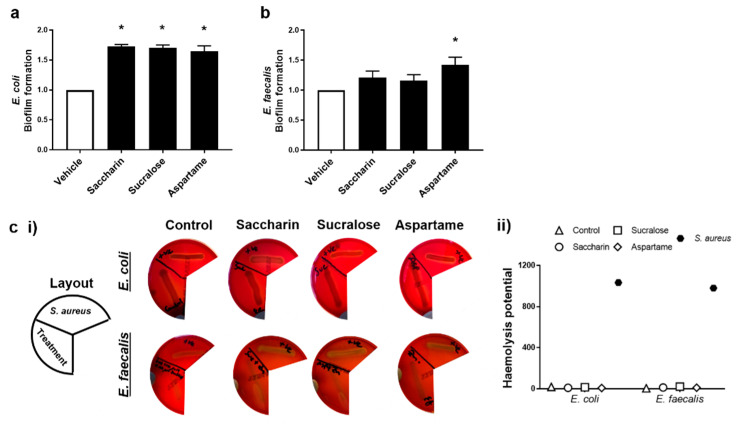
Artificial sweeteners differentially increase biofilm formation, but not haemolytic activity, in the two model gut bacteria. *E*. *coli* (panel **a**) and *E. faecalis* (panel **b**) were exposed to artificial sweeteners, saccharin, sucralose, and aspartame (100 µM) for 24 h (panels **a**,**b**) or 48 h (panel **c**) and biofilm formation or haemolysis assay, respectively, were performed. For the haemolysis assay, S. *aureus* was used as a positive control to show haemolysin release (panel **c**) using representative images (panel **c** (**i**)) and semi-quantification (panel **c** (**ii**)) of haemolysis. Specific bacteria for each study are included in underlined text. Data are presented as mean ± S.E.M for panel a and b and as mean only for panel c. * *p* < 0.05 versus vehicle for biofilm formation.

**Figure 3 ijms-22-05228-f003:**
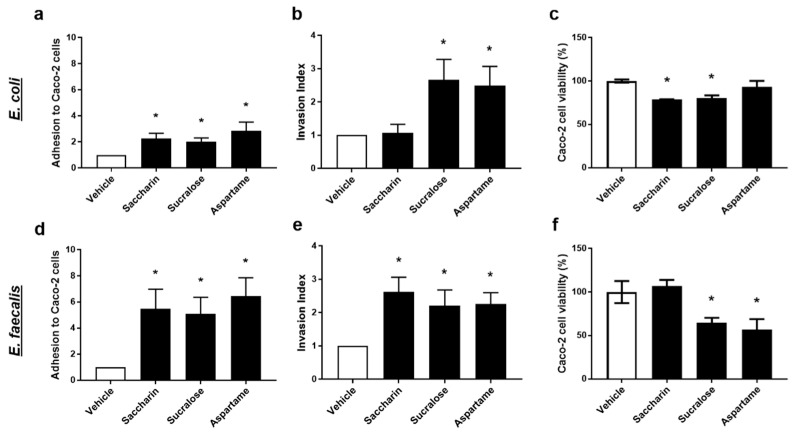
Artificial sweeteners significantly disrupt the interaction between model gut bacterial and intestinal epithelial cells. *E. coli* (panels **a**–**c**) and *E. faecalis* (panels **d**–**f**) were exposed to artificial sweeteners, saccharin, sucralose, and aspartame (100 µM), for 24 h. Adhesion of bacteria to Caco-2 cells (panels **a**,**d**) and invasion of bacteria into Caco-2 cells (panels **b**,**e**) were measured following incubation with Caco-2 cells for 1 h. Cytotoxic effects of bacterial soluble factors were assessed by culturing bacterial supernatant with Caco-2 cells for 24 h (panels **c**,**f**). Specific bacteria for each study are included in underlined text. Data are presented as mean ± S.E.M. * *p* < 0.05 versus vehicle.

**Figure 4 ijms-22-05228-f004:**
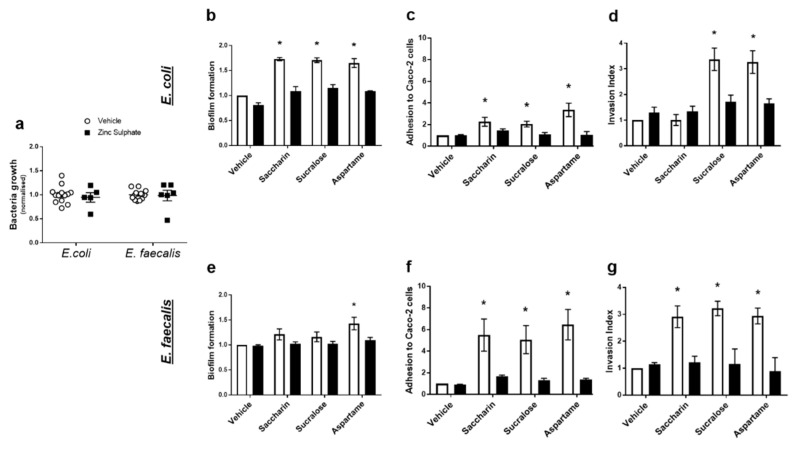
Artificial sweeteners impact model gut bacteria through a taste sensing mechanism**.**
*E. coli* and *E. faecalis* were exposed to zinc sulphate (100 µM) for 24 h and growth was measured (panel **a**). Alternatively, *E. coli* (panels **b**–**d**) and *E. faecalis* (panels **e**–**g**) were exposed to artificial sweeteners, saccharin, sucralose, and aspartame (100 µM), for 24 h in the presence or absence of zinc sulphate (100 µM). Subsequent measurements were made to assess biofilm formation (panels **b**,**e**), adhesion to (panels **c**,**f**) and invasion of (panels **d**,**g**) Caco-2 cells. Specific bacteria for each study are included in underlined text. Data are presented as mean ± S.E.M. * *p* < 0.05 versus vehicle for sweeteners.

## Data Availability

Data sharing not applicable.

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
