# Peer review of "Artificial Sweeteners Negatively Regulate Pathogenic Characteristics of Two Model Gut Bacteria, E. coli and E. faecalis"

_ijms, 2021, doi:10.3390/ijms22105228_

Round 1

Reviewer 1 Report

In this study, Shil et al. investigated in vitro the effects of 3 artificial sweeteners (AS, saccharin, sucralose, and aspartame) on 2 model gut bacteria, E. coli NCT and E. faecalis. They first determined the doses which did not impact bacteria cell growth. Then they assessed the effects of AS on biofilm formation, haemolytic activity, ability to adhere to, invade and kill Caco-2 intestinal epithelial cells. They also assessed the mechanisms involved in these effects, and confirmed the involvement of taste sensing by using zinc sulphate as inhibitor of sweet taste sensing. This manuscript is clear. The methods are accurate and very well described. The results are new.

I have a few minor comments.

The authors should try to increase the size of the police in the figures.

In figure 1a, the statistics do not appear, whereas it is mentioned in the results and in the legend that the difference is significant.

Unfortunately, the lines are not numbered but I would like to mention little mistakes:

“aspartame (figure 2b)” should be replaced by “…2a”

“To form a biofilm (Figure 2c)” should be replaced by “…2b”

In the legend of figure 2, “haemolysis assay or biofilm formation assay” should be inversed.

“Epithelial cell line, Caco-Studies”: something is missing.

“Epithelial cells (Figure 3a and c)” should be replaced by “…3a and d”

In the legend of figure 3, “panels b and d” should be replaced by “…b and e”  

In the description of invasion assay, something is missing:

“5% COThe cell”

“Taken as Each assay”

Author Response

Reviewer 1

In this study, Shil et al. investigated in vitro the effects of 3 artificial sweeteners (AS, saccharin, sucralose, and aspartame) on 2 model gut bacteria, E. coli NCT and E. faecalis. They first determined the doses which did not impact bacteria cell growth. Then they assessed the effects of AS on biofilm formation, haemolytic activity, ability to adhere to, invade and kill Caco-2 intestinal epithelial cells. They also assessed the mechanisms involved in these effects and confirmed the involvement of taste sensing by using zinc sulphate as inhibitor of sweet taste sensing. This manuscript is clear. The methods are accurate and very well described. The results are new.

We thank reviewer 1 for these comments and have addressed the specific points and queries as below:

  • The authors should try to increase the size of the police (text? Font?) in the figures.
  • In figure 1a, the statistics do not appear, whereas it is mentioned in the results and in the legend that the difference is significant.

Thank you for points 1) and 2) – it appears that there was a technical issue in transferring the figures to the Word doc template. This should now have been resolved.

  • Unfortunately, the lines are not numbered but I would like to mention little mistakes:
  • “aspartame (figure 2b)” should be replaced by “…2a”
  • “To form a biofilm (Figure 2c)” should be replaced by “…2b”
  • In the legend of figure 2, “haemolysis assay or biofilm formation assay” should be inversed.
  • “Epithelial cell line, Caco-Studies”: something is missing.
  • “Epithelial cells (Figure 3a and c)” should be replaced by “…3a and d”
  • In the legend of figure 3, “panels b and d” should be replaced by “…b and e”  
  • In the description of invasion assay, something is missing:

“5% COThe cell”

“Taken as Each assay”

Thank you for these comments. They have been amended and corrected as suggested in the manuscript.

Reviewer 2 Report

Authors Shil and Chichger submitted their manuscript entitled "Artificial sweeteners negatively regulate pathogenic characteristics of two model gut bacteria, E. coli NCT and E. faecalis” to IJMS. The manuscript covers an interesting and actual topic. It is readable and thoroughly written. Artificial sweeteners (AS) are usually studied from the point of the host's metabolism. The authors studied the use of AS from the point of microbiota – on the defined model of Caco-2 cell lines as host cells and E. coli and E. faecalis as intestinal microbiota representatives.

I have several recommendations and notices. Hopefully, it could help to improve the manuscript.

L11: Microbiota instead of microflora is preferred at least within the last decade. Please, judge possible replace these terms through the manuscript.

L15: Here should be "Caco-2 cell line" or " Caco-2 cells".

L16: "AS, saccharin" should probably be changed to "AS saccharin".

L26-27: This information is old. The ratio of bacterial cells : body cells is approx. 1 : 1. Please, add to the appropriate reference. If you would like to keep the ratio 10 : 1, please also add the newer opinion dealing with the ratio. https://doi.org/10.1016/j.cell.2016.01.013

L37-38: Would it not be suitable to mention the production of B group vitamins and vitamin K by microbiota that is essential for the appropriate development of the host?

L67, 68, 75, etc.: You introduced the abbreviation AS (artificial sweeteners) in the abstract. If abbreviated, it would be suitable to use this abbreviation through the whole text of the manuscript.

L98-99: The last sentence of the paragraph would be suitable for the conclusion but not for the introduction. Please, remove it.

L100-101: I know that E. coli is notoriously known. However, the first use should be Escherichia coli and later E. coli. The same in the case of Enterococcus faecalis.

L127: Figure 1. Please, increase font size in the description of the x- and y-axes as well as legends. It is hardly readable.

L137-138: It is not the description of results but discussion. Please, remove it or rephrase the sentence.

L148 Figure a, b, and ii: Please, increase the font in axes x and y and figure legend (ii) to be easily readable.

L155-159: It should be a sober description of results without any explanations, speculations, etc. Please remove or rephrase the paragraph.

L169 Figure 3: Please, increase the font and asterisks to be readable and describe which characteristics are depicted - probably mean + SD or SEM.

The vehicle did not show any variability in figs a, b, d, e?

L174-175: Please did not explain your intention in Results

L186-190: Please do not explain. It is not a discussion. Please, rephrase the sentences.

L204 Figure 4. Please, increase fonts and include information which characteristics are depicted.

L210-214: It is not a description of results but a discussion. Please, remove the text.

L254-255: Please, modify "Pathogenic E. coli" to "Some pathogenic E. coli", "Some E. coli pathotypes", etc. E.g., atypical EPEC (aEPEC) did not have bfp genes.

L320: Here should be suitable to add some text, e.g., as you have on lines 97-99.

L335-342: The methods should be more detailed to be possible to follow them by other researchers. Please, precise it.

L341-342: What does it mean 1% antibiotic? Please, express their concentration in appropriate units.

L383-384: Please, add which media did you use in the assay.

L400: Please, add which medium did you use.

L418-419: Please, add the used medium.

L422: It would be necessary to know a rotor radius to imagine and follow the condition of your centrifugation force. Please, express it in a centrifugation force as x g to be re-countable for any other centrifuge.

L430-432: You used parametric and non-parametric tests. It is not clear in which case which of them was used. Moreover, I would like to note that characteristics of parametric tests are mean and SD or SEM, but in the case of non-parametric tests, they are, e.g., a median and range. Please, distinguish the used tests and depicted characteristics. Please, add which test you used to evaluate the normality of value distribution to choose the parametric or non-parametric tests.

Author Response

Reviewer 2:

Authors Shil and Chichger submitted their manuscript entitled "Artificial sweeteners negatively regulate pathogenic characteristics of two model gut bacteria, E. coli NCT and E. faecalis” to IJMS. The manuscript covers an interesting and actual topic. It is readable and thoroughly written. Artificial sweeteners (AS) are usually studied from the point of the host's metabolism. The authors studied the use of AS from the point of microbiota – on the defined model of Caco-2 cell lines as host cells and E. coli and E. faecalis as intestinal microbiota representatives. I have several recommendations and notices. Hopefully, it could help to improve the manuscript.

We thank reviewer 2 for these comments and have addressed the specific points and queries as below:

  • L11: Microbiota instead of microflora is preferred at least within the last decade. Please, judge possible replace these terms through the manuscript.
  • L15: Here should be "Caco-2 cell line" or " Caco-2 cells".
  • L16: "AS, saccharin" should probably be changed to "AS saccharin".

Thank you for these comments. They have been amended and corrected as suggested in the manuscript.

  • L26-27: This information is old. The ratio of bacterial cells : body cells is approx. 1 : 1. Please, add to the appropriate reference. If you would like to keep the ratio 10 : 1, please also add the newer opinion dealing with the ratio. https://doi.org/10.1016/j.cell.2016.01.013

We thank the reviewer for this point and appreciate the newer literature. We have now adjusted the first statement in the paper to allow for this.

  • L37-38: Would it not be suitable to mention the production of B group vitamins and vitamin K by microbiota that is essential for the appropriate development of the host?

We thank the reviewer for this point and have raised it in the second paragraph of the introduction.

  • L98-99: The last sentence of the paragraph would be suitable for the conclusion but not for the introduction. Please, remove it.

We appreciate this comment from the reviewer. Given that there is no set publication style for IJMS, we chose to leave this sentence in the Introduction to give context and meaning to the paper. We also believe it is important for readers to understand this impact before reading the Results section and feel this gives better flow the paper.

  • L100-101: I know that E. coli is notoriously known. However, the first use should be Escherichia coli and later E. coli. The same in the case of Enterococcus faecalis.

We appreciate this point and have given the full name for each in the Introduction.

  • L127: Figure 1. Please, increase font size in the description of the x- and y-axes as well as legends. It is hardly readable.

Thank you for this point – it appears to be a technical issue in transferring figures to Word. This is now resolved.

  • L137-138: It is not the description of results but discussion. Please, remove it or rephrase the sentence.

We appreciate this comment and have adjusted the sentence to focus on the results.

  • L148 Figure a, b, and ii: Please, increase the font in axes x and y and figure legend (ii) to be easily readable.

Thank you for this point – it appears to be a technical issue in transferring figures to Word. This is now resolved.

  • L155-159: It should be a sober description of results without any explanations, speculations, etc. Please remove or rephrase the paragraph.

As for our response to comment 6), we appreciate this comment from the reviewer however, we chose to leave this sentence in to give context, meaning and flow to the paper and provide depth to the results section.

  • L169 Figure 3: Please, increase the font and asterisks to be readable and describe which characteristics are depicted - probably mean + SD or SEM.

Thank you for this point – the font size and asterisk appears to be a technical issue in transferring figures to Word. This is now resolved. We have also added a statement to each legend clarifying that data is mean ± standard error mean with the exception of Figure 2 c ii.

  • The vehicle did not show any variability in figs a, b, d, e?

This is correct as we normalise data to the vehicle (denoted as 1). This has been described in the methods for each assay.

  • L174-175: Please did not explain your intention in Results
  • L186-190: Please do not explain. It is not a discussion. Please, rephrase the sentences.

As for our response to comment 6) and 11), we appreciate this comment from the reviewer however, we chose to leave this sentence in to give context, meaning and flow to the paper and provide depth to the results section.

  • L204 Figure 4. Please, increase fonts and include information which characteristics are depicted.

Thank you for this point – as for other figures, this is now resolved.

  • L210-214: It is not a description of results but a discussion. Please, remove the text.

As for our response to comment 6), 11), 14) and 15), we appreciate this comment from the reviewer however, we chose to leave this sentence in to give context, meaning and flow to the paper and provide depth to the results section.

  • L254-255: Please, modify "Pathogenic E. coli" to "Some pathogenic E. coli", "Some E. coli pathotypes", etc. E.g., atypical EPEC (aEPEC) did not have bfp genes.

Thank you for this comment, we have adjusted as suggested.

  • L320: Here should be suitable to add some text, e.g., as you have on lines 97-99.

The last sentence of the paragraph would be suitable for the conclusion but not for the introduction. Please, remove it.

We appreciate this comment and have addressed it in our response for comment 6), 11), 14), 15) and 17).

  • L335-342: The methods should be more detailed to be possible to follow them by other researchers. Please, precise it.

We thank the reviewer for this point. Having reviewed similar papers in IJMS and other journals, we feel the level of detail in the Methods is sufficient for use by other researchers. This is especially relevant as we refer to previous studies which originally developed the technique.

  • L341-342: What does it mean 1% antibiotic? Please, express their concentration in appropriate units.

We appreciate the comment and have included the precise concentration for the antibiotics in the text

  • L383-384: Please, add which media did you use in the assay.

  • L400: Please, add which medium did you use.
  • L418-419: Please, add the used medium.

To avoid repetition, the media used in all bacterial and mammalian cell culture assays was included in the Methods in paragraph 4.2: ‘Brain Heart Infusion agar and Nutrient Agar were used as solid, and Brain Heart Infusion and Nutrient broth were used as liquid media for E. faecalis and E. coli, respectively, as guided by the supplier.’

  • L422: It would be necessary to know a rotor radius to imagine and follow the condition of your centrifugation force. Please, express it in a centrifugation force as x g to be re-countable for any other centrifuge.

The centrifugation force (2683 x g) has now been included in the methods section.

  • L430-432: You used parametric and non-parametric tests. It is not clear in which case which of them was used. Moreover, I would like to note that characteristics of parametric tests are mean and SD or SEM, but in the case of non-parametric tests, they are, e.g., a median and range. Please, distinguish the used tests and depicted characteristics. Please, add which test you used to evaluate the normality of value distribution to choose the parametric or non-parametric tests.

Thank you for this comment. We made an error with the number of statistical tests included, based on other studies performed on the project, and have now corrected the Methods section accordingly (in section 4.9)

Round 2

Reviewer 2 Report

The manuscript was partially improved. However, the methods are very roughly described to be possible to make imagination or follow them. The references to papers with detail methodical descriptions are very often confusing. If you search from the manuscript to the referred paper 1, instead of the description of the method, it refers to paper 2 that refers to paper 3 … It is sometimes laborious to find which method was used. It is the reason why at least a brief description of the used method should be added. The problematic methods I noticed in my first review.

It has still been confused description of results and discussion. It should be corrected as described in the previous review.

L140-143: It is not the description of results but the explanation of the experiments. It should be in the discussion but not in the results.

L157-158: It is not the description of results but the explanation suitable for discussion.

L167-170: It is not the description of results but the explanation suitable for discussion.

L187-190: It is not the description of results but the explanation suitable for discussion. Moreover, the last sentence is suitable for the conclusion but not for the description of the results.

L288: 100 U/0.1 mg is not the description of concentration but the description of amount. Please, correct it.

L355-355: Non-parametric test was excluded in version 2 of the manuscript. It is not obvious how you evaluate the normality of distribution to decide to use the parametric test only. Triplicates were used. Did you calculate the normality of the distribution from three values only, or were experiments repeated several times to obtain a sufficient number of values that allow calculation of their distribution? Please, clarify it.

Figures: All results are not presented as mean ± S.E.M. but as mean + S.E.M - see black columns. Please correct texts or figures.

Author Response

Response to Reviewer 2 comments

  • The manuscript was partially improved. However, the methods are very roughly described to be possible to make imagination or follow them. The references to papers with detail methodical descriptions are very often confusing. If you search from the manuscript to the referred paper 1, instead of the description of the method, it refers to paper 2 that refers to paper 3 … It is sometimes laborious to find which method was used. It is the reason why at least a brief description of the used method should be added. The problematic methods I noticed in my first review.

We thank the reviewer for these further comments and have now updated the manuscript with further detail in the methods of the adhesion and invasion assay.

  • It has still been confused description of results and discussion. It should be corrected as described in the previous review.
  • L140-143: It is not the description of results but the explanation of the experiments. It should be in the discussion but not in the results.
  • L157-158: It is not the description of results but the explanation suitable for discussion.
  • L167-170: It is not the description of results but the explanation suitable for discussion.
  • L187-190: It is not the description of results but the explanation suitable for discussion. Moreover, the last sentence is suitable for the conclusion but not for the description of the results.

Whilst we understand the reviewer’s comment, we respectfully disagree that there is confusion in the description of the results and the discussion. In the Results section, we describe findings from our data and provide a couple of statements, at the beginning and end of each results paragraph, to provide context and flow to the reader. This is very important to be clear on why each study was performed and is vital in providing structure to the paper. We have reviewed many other journals and papers within IJMS and find this same style is used. We therefore have not altered the results section.

  • L288: 100 U/0.1 mg is not the description of concentration but the description of amount. Please, correct it.

Thank you for this comment and apologies for the error. This has now been corrected.

  • L355-355: Non-parametric test was excluded in version 2 of the manuscript. It is not obvious how you evaluate the normality of distribution to decide to use the parametric test only. Triplicates were used. Did you calculate the normality of the distribution from three values only, or were experiments repeated several times to obtain a sufficient number of values that allow calculation of their distribution? Please, clarify it.

We appreciate the reviewer’s comment and have added detail to the statistical analysis section to outline how triplicates were used, the n number used for the studies, and the post-hoc comparison tests utilised.

  • Figures: All results are not presented as mean ± S.E.M. but as mean + S.E.M - see black columns. Please correct texts or figures.

We thank the reviewer for this comment however data is presented +/- SEM. The black columns simply block the ability to view the negative SEM bar therefore we have left this statement in the figure legend.